# Membrane Binding of Neuronal Calcium Sensor-1: Highly Specific Interaction with Phosphatidylinositol-3-Phosphate

**DOI:** 10.3390/biom10020164

**Published:** 2020-01-21

**Authors:** Viktoriia E. Baksheeva, Ekaterina L. Nemashkalova, Alexander M. Firsov, Arthur O. Zalevsky, Vasily I. Vladimirov, Natalia K. Tikhomirova, Pavel P. Philippov, Andrey A. Zamyatnin, Dmitry V. Zinchenko, Yuri N. Antonenko, Sergey E. Permyakov, Evgeni Yu. Zernii

**Affiliations:** 1Belozersky Institute of Physico-Chemical Biology, Lomonosov Moscow State University, 119992 Moscow, Russia; vibaksheeva@gmail.com (V.E.B.); firsov@belozersky.msu.ru (A.M.F.); tikhomir@belozersky.msu.ru (N.K.T.); ppph@belozersky.msu.ru (P.P.P.); antonen@belozersky.msu.ru (Y.N.A.); 2Institute for Biological Instrumentation of the Russian Academy of Sciences, Pushchino, 142290 Moscow Region, Russia; elnemashkalova@gmail.com (E.L.N.); permyakov.s@gmail.com (S.E.P.); 3Faculty of Bioengineering and Bioinformatics, Lomonosov Moscow State University, 119992 Moscow, Russia; aozalevsky@fbb.msu.ru; 4Shemyakin and Ovchinnikov Institute of Bioorganic Chemistry of the Russian Academy of Sciences, 117997 Moscow, Russia; 5Institute of Molecular Medicine, Sechenov First Moscow State Medical University, 119991 Moscow, Russia; 6Branch of Shemyakin and Ovchinnikov Institute of Bioorganic Chemistry of the Russian Academy of Sciences in Pushchino, Pushchino, 142290 Moscow Region, Russia; vladimirov@bibch.ru (V.I.V.); zdv@bibch.ru (D.V.Z.)

**Keywords:** neuronal calcium sensors, neuronal calcium sensor-1, NCS-1, membrane binding, N-terminal myristoylation, myristoyl group, phospholipid-binding proteins, phosphoinositides, phosphatidylinositol-3-phosphate, PI3P

## Abstract

Neuronal calcium sensors are a family of N-terminally myristoylated membrane-binding proteins possessing a different intracellular localization and thereby targeting unique signaling partner(s). Apart from the myristoyl group, the membrane attachment of these proteins may be modulated by their N-terminal positively charged residues responsible for specific recognition of the membrane components. Here, we examined the interaction of neuronal calcium sensor-1 (NCS-1) with natural membranes of different lipid composition as well as individual phospholipids in form of multilamellar liposomes or immobilized monolayers and characterized the role of myristoyl group and N-terminal lysine residues in membrane binding and phospholipid preference of the protein. NCS-1 binds to photoreceptor and hippocampal membranes in a Ca^2+^-independent manner and the binding is attenuated in the absence of myristoyl group. Meanwhile, the interaction with photoreceptor membranes is less dependent on myristoylation and more sensitive to replacement of K3, K7, and/or K9 of NCS-1 by glutamic acid, reflecting affinity of the protein to negatively charged phospholipids. Consistently, among the major phospholipids, NCS-1 preferentially interacts with phosphatidylserine and phosphatidylinositol with micromolar affinity and the interaction with the former is inhibited upon mutating of N-terminal lysines of the protein. Remarkably, NCS-1 demonstrates pronounced specific binding to phosphoinositides with high preference for phosphatidylinositol-3-phosphate. The binding does not depend on myristoylation and, unexpectedly, is not sensitive to the charge inversion mutations. Instead, phosphatidylinositol-3-phosphate can be recognized by a specific site located in the N-terminal region of the protein. These data provide important novel insights into the general mechanism of membrane binding of NCS-1 and its targeting to specific phospholipids ensuring involvement of the protein in phosphoinositide-regulated signaling pathways.

## 1. Introduction

Many signaling pathways in neurons involve the recruitment of proteins to the membrane surface in response to stimuli. Membrane targeting of proteins is achieved by a variety of specific lipid-binding domains as well as co-translational modifications, such as N-terminal myristoylation [1,2]. The latter strategy is used by neuronal calcium sensors (NCSs), a family of calcium-binding EF-hand proteins playing different signaling roles in the nervous system (for review, see [3]). In the majority of Ca^2+^-free NCS proteins, the myristoyl group is buried inside their so-called hydrophobic pocket. In response to Ca^2+^ binding, the residues forming this pocket and the myristoyl group can be exposed to the surface of the protein molecule. This mechanism known as the Ca^2+^-myristoyl switch allows NCSs to reversibly interact with cellular membranes via the myristoyl group and specifically regulate their targets (which are mostly integral or membrane-associated proteins) via binding them by amino acids of the hydrophobic pocket [4,5,6,7]. Notably, while all NCSs have similar core structure, multiple members of the family can be expressed simultaneously in the same neuron without overlapping of their functions. This differentiation is, in part, achieved by their diverse subcellular localization [8,9,10,11]. Thus, apart from the myristoyl group, each NCS might possess unique structural elements, which contain additional targeting information, selectively linking them to membranes based on their lipid composition.

Indeed, growing evidence indicates that membrane localization of myristoylated proteins can be affected by their N-terminal basic amino acids, located in the proximity of the fatty acid moiety [8,12,13,14]. While NCSs do not possess conventional phospholipid-binding domains [1], amino acids of the N-terminal region of these proteins contribute to their translocation either to the plasma membrane or to intracellular membrane structures, such as the Golgi complex. For example, the first 14 residues of hippocalcin target the protein to membranes enriched in phosphatidylinositol-4,5-bisphosphate (PI(4,5)P_2_) [8]. Furthermore, the affinity of hippocalcin to this phospholipid is strong enough to displace the PH domain of phospholipase C, the known marker of PI(4,5)P_2_, from the Golgi apparatus. Another NCS protein, recoverin, has an alternative arrangement of the N-terminal charged residues. NMR studies revealed that recoverin localizes on the surface of the phospholipid membrane with an array of lysine and arginine residues flanking the myristoyl group facing downward towards the membrane, where they can form contacts with negatively charged head groups of the phospholipids [13]. While no phospholipid preference has been described for recoverin yet, its Ca^2+^-induced activation translocates the protein to the membranes of photoreceptor discs, which are specifically enriched in phosphatidylserine and extremely low on phosphatidylinositol in comparison to general neuron membranes and especially the Golgi membranes [15,16,17,18]. Unlike both these NCSs, Kv-channel interacting protein-1 (KChIP1) lacks the positively charged residues in its N-terminal segment, and is not attracted to specific phospholipids, but rather to punctate structures on the plasma membrane or the endoplasmic reticulum, where it co-localizes with its target Kv4 ion channels [8,19]. Overall, the N-terminal segment of NCSs appears to play a key role in the distinct subcellular localization of these proteins.

Neuronal calcium sensor-1 (NCS-1) is a ubiquitous NCS protein belonging to the frequenin branch of the family. It is involved in the regulation of neurotransmission, reception and synaptic plasticity, as well as growth and survival of neurons (for review, see [20]). Among NCSs, NCS-1 is characterized by the largest number of known regulatory targets. Like hippocalcin, NCS-1 predominantly localizes in the perinuclear region of neurons, where it is associated with the trans-Golgi network, as well as on plasma membrane and in pre- and post-synaptic structures [21,22]. NCS-1 is also present in the retina, where it is mainly found in the synaptic layers and photoreceptor inner segments [9]. Our recent studies indicate that NCS-1 can also be present in photoreceptor outer segments, where it co-localizes with scaffold protein caveolin-1 and can be involved in photoreceptor-specific signaling pathways [23,24]. Finally, NCS-1 is found outside the nervous system, in neuroendocrine and immune cells, where it is involved in the regulation of secretion via its interaction with phosphatidylinositol-4-kinases [25,26]. This diversity of functions seems to clash with its unique mode of membrane binding. According to the accumulated data, NCS-1, unlike the Ca^2+^-myristoyl switch NCS proteins, maintains its localization under Ca^2+^-free conditions via constant exposure of its myristoyl group [10,24,27,28,29]. The irreversibility of its membrane association points to the importance of precise intracellular targeting of NCS-1 based on the lipid and protein composition of the membrane. Previously, by analogy with hippocalcin, it was hypothesized that membrane association and phospholipids specificity of NCS-1 can be determined by its N-terminal domain and particularly the lysine residues in positions 3, 7 and 9, conserved between these proteins [8,19,28]. However, no direct experiments confirming this hypothesis have been conducted.

In this study, we explored NCS-1 binding to two types of neuronal membranes with different phospholipid composition, namely photoreceptor membranes (phosphatidylserine (PS) >> phosphatidylinositol (PI)) and hippocampal membranes (PS ≈ PI), as well as to individual phospholipids in form of multilamellar liposomes or immobilized monolayers. Particularly, we assessed the role of the N-terminal myristoyl group and adjacent positively charged residues of the protein in its membrane binding and phospholipid preference. We found that NCS-1 binds to neuronal membranes of both types in a Ca^2+^-independent manner, but the binding to photoreceptor membranes is less dependent on myristoylation and more prominently attenuated upon replacement of K3, K7, and/or K9 of the protein by glutamic acid indicating affinity of the protein to negatively charged phospholipids. Consistently, among the major phospholipids, unmyristoylated NCS-1 interacts with phosphatidylserine and phosphatidylinositol and the interaction with the former is inhibited upon mutating of N-terminal lysines of the protein. Remarkably, NCS-1 displays a strong affinity to low abundance phosphoinositides, especially phosphatidylinositol-3-phosphate (PI3P). This feature of the protein is insensitive to the presence of myristoyl group or charge inversions in its the N-terminus. Instead, PI3P is likely recognized by a novel specific site located in the N-terminal region of NCS-1. Thus, the N-terminus of NCS-1 ensures both general membrane anchoring of the protein by providing a myristoyl group and positively charged residues and its highly specific binding to PI3P by forming the respective site. In the aggregate, these findings provide novel insights into the mechanism of membrane binding of NCS-1 and confirm its involvement in phosphoinositide (especially PI3P)-regulated signaling pathways.

## 2. Materials and Methods

### 2.1. Materials

Oligonucleotide primers and reagents for cloning were from Evrogen (Moscow, Russia). Chromatography columns were from GE Healthcare Life Sciences (Marlborough, MA, USA). Primary antibodies against NCS-1 were produced by rabbit immunization and purified according to the previously published procedure [30]. Secondary antibodies were from Santa Cruz Biotechnology, Dallas, TX, USA (sc-2030). Other chemicals were from Sigma-Aldrich (St. Louis, MO, USA), Amresco (Solon, OH, USA) and PanEco (Moscow, Russia) and were at least of reagent grade.

### 2.2. Cloning of NCS-1 N-Terminal Mutants

The genetic construct encoding NCS-1^WT^ gene was previously created in our laboratory [24]. Point mutations were introduced into the gene by PCR, using antisense primer 5′-GGGAAGCTTATACCAGCCCGTCGTAGAGGG-3′ and a set of sense primers with specific lysine codons substituted with glutamic acid codons: 5′-GGGCATATGGGGGAATCCAACAGC-3′ (K3E); 5′-GGGCATATGGGGAAATCCAACAGCGAGTTG-3′ (K7E); 5′-GGGCATATGGGGAAATCCAACAGCAAGTTGGAGCCTG-3′ (K9E). Double substitution mutants were produced by subsequent PCRs on basis of K7E and K9E mutants and K3,7,9E mutant was created on basis of K7,9E mutant, using the corresponding set of primers. The amplified mutant genes were subcloned into pET22b+ expression vector between NdeI and HindIII endonuclease restriction sites. All constructs were verified by automated sequencing.

### 2.3. Purification and Characterization of Myristoylated and Unmyristoylated Recombinant NCS-1 and Its Mutant Forms

Competent *Escherichia coli* cells (strain BL-21(DE3)CodonPlus) were co-transformed with vectors encoding NCS-1^WT^ and PBB131 plasmid containing the myristoyl-CoA:protein N-myristoyltransferase (NMT) gene from *Saccharomyces cerevisiae*. Co-expression of NCS-1 variants with NMT and subsequent purification of the myristoylated proteins from bacterial lysate was carried out as previously described for NCS-1^WT^ [31]. Myristoylation levels of the produced proteins were determined by analytical high-performance liquid chromatography (HPLC) using a reversed-phase column (Phenomenex Luna C18(2)) [32]. Unmyristoylated NCS-1^WT^ and its N-terminal mutants were expressed in the absence of the NMT gene and purified in the same manner. The concentration of NCS-1 forms was determined spectrophotometrically at 280 nm using the molar extinction coefficient of 21,430 M^−1^. The thermal stability of all analyzed proteins was characterized by monitoring the temperature dependence of the tryptophan fluorescence spectrum maximum position (λ_max_) in the presence of either 1 mM CaCl_2_ or 1 mM MgCl_2_ and 1 mM ethylene glycol tetraacetic acid (EGTA) as described in [24].

### 2.4. Preparation of Urea-Washed Photoreceptor and Hippocampal Membranes

Rod outer segment (ROS) preparations were obtained from frozen retinae in dim red light by the standard protocol [33]. Urea-washed photoreceptor (ROS) membranes were prepared as described in [34]. Hippocampal membranes were prepared as described in [35] with slight modifications. Briefly, bovine hippocampus was ground on ice with a glass-teflon homogenizer in 30 volumes of 50 mM Tris-HCl buffer (pH 8.0), containing 1 mM ethylenediaminetetraacetic acid (EDTA), 1 mM phenylmethylsulfonyl fluoride and 10% sucrose. All following centrifugation steps were performed at 4 °C. The suspension was centrifuged at 1000 g for 10 min, after which the pellet was discarded and the supernatant was centrifuged at 22,000 g for 30 min. Photoreceptor and hippocampal membrane pellets were next resuspended in 10 volumes of 20 mM Tris-HCl buffer (pH 8.0), 5 mM EDTA and 5 M urea, kept on ice for 10 min and centrifuged at 150,000 g for 40 min. The pellets were washed by 20 mM Tris-HCl buffer (pH 8.0) containing 1 mM dithiothreitol (DTT) three times, resuspended in 1 volume of the same buffer, aliquoted and stored at −20 °C.

### 2.5. Membrane Binding Assay

NCS-1 binding to urea-washed membranes was performed by equilibrium centrifugation assay, as described in [31,36] with modifications. Briefly, 5 µL of the suspension of photoreceptor or hippocampal membranes were mixed with 40 µM NCS-1 in 20 mM Tris-HCl buffer (pH 8.0), 150 mM NaCl, 1 mM DTT, 20 mM MgCl_2_ and either 2 mM CaCl_2_ or 2 mM EGTA in the total volume of 50 µL. After agitation in a thermostatic shaker for 30 min (37 °C, 1000 rpm) the membrane pellet was collected by centrifugation (24,000 g, 15 min) and dissolved in 50 µL of SDS-PAGE buffer. Membrane-bound NCS-1 was visualized by SDS-PAGE with Coomassie Brilliant Blue staining as well as by Western blotting and its weight fractions were quantitated using GelAnalyzer software for densitometric analysis [37].

### 2.6. Analysis of NCS-1 Binding to Liposomes by Fluorescence Correlation Spectroscopy

Myristoylated and unmyristoylated NCS-1 was conjugated with sulfo-cyanine-3 maleimide fluorescent tag (Lumiprobe, Hannover, Germany) according to the manufacturer instructions. The conjugate was separated by the unbound dye by chromatography using Sephadex 15 column.

Multilamellar liposomes were prepared by evaporation under a stream of nitrogen of a 2% solution of brain PS, egg yolk PC, dimyristoyl-PE, and brain PI (all lipids were from Sigma-Aldrich) in chloroform/methanol (2:1) followed by hydration with a buffer solution containing 150 mM NaCl, 10 NaH_2_PO_4_ (pH 7.0). The mixture was vortexed and passed through a cycle of freezing and thawing. Vesicles were visible in a microscope and varied in size with some preference around 3–5 µm.

The setup for fluorescence correlation spectroscopy (FCS) was described previously [38]. Briefly, fluorescence excitation and detection were performed using an Nd:YAG solid-state laser with a 532-nm beam attached to an Olympus IMT-2 epifluorescent inverted microscope with a 40 ×, NA 1.2 water immersion objective (Carl Zeiss, Oberkochen, Germany). After passing through an appropriate dichroic beam splitter and a long-pass filter the fluorescence light was imaged onto a 50 μm core fiber coupled to an avalanche photodiode (SPCM-AQR-13-FC, PerkinElmer Optoelectronics, Waltham, MA, USA). The output was sent to a personal computer via a fast interface card (Flex02-01D/C, Correlator.com, Bridgewater, NJ, USA). The fluorescence was recorded from the confocal volume located at about 50 μm above the coverslip surface carrying 50 μL of the sample solution, containing 100 nM NCS-1 in buffer composed of 20 mM Tris-HCl (pH 8.0), 100 mM NaCl and either 1 mM CaCl_2_ or 1 mM EGTA. Data acquisition time was 30 s, during which the sample was stirred by a paddle-shaped 3 mm plastic bar rotated at 600 rpm. The setup was calibrated by recording the autocorrelation function of the fluorescence of Rhodamine 6G solution without stirring. The correlated fluorescence emission signals were fitted to the three-dimensional autocorrelation function [39] with τ_D_ being the characteristic correlation time during which a molecule resides in the observation volume of radius ω and length z_0_, given by τ_D_ = ω^2^/4D, where D is the diffusion coefficient, N is the mean number of fluorescent particles in the confocal volume. We used the software supplied with the Flex02-01D/C interface card for fitting procedures, as well as the SigmaStat software (Systat Software, San Jose, CA, USA). The correlation time τ_D_ was calculated from Equation (1). The assumed diffusion coefficient of the Rhodamine 6G dye was 4.26 × 10^−6^ cm^2^/s [40]. Thus, the value of the confocal radius ω = 0.55 μm was obtained. NCS-1 with fluorescent tag exhibited slower diffusion compared to that of Rhodamine 6G resulting in τ_D_ of about 1000 µs corresponding to the diffusion coefficient of about 85 µm^2^/s.
(1)$$G\left(\tau \right)=1+\frac{1}{N}\left(\frac{1}{1+\frac{\tau }{\tau_{D}}}\right)\left(\frac{1}{\sqrt{1+\frac{\omega^{2}\tau }{z_{0}^{2}\tau_{D}}}}\right)$$

The protein binding to liposomes was estimated from the number of peaks of the fluorescence exceeding a certain threshold as described in [41] (Figure A1). Fluorescence traces with the sampling time of 25 μs were analyzed using WinEDR Strathclyde Electrophysiology Software designed by J. Dempster (University of Strathclyde, Glasgow, UK) or another program with an algorithm developed by Wladas Kozlovsky (Belozersky Institute, Moscow State University, Moscow, Russia). The software, originally designed for the single-channel analysis of electrophysiological data, enables one to count the number of peaks (n(F > F_0_)) of the FCS signal having amplitudes higher than the defined value (F_0_). Here, the threshold of 0.4 MHz was used. Where possible, dissociation constants were assessed by titration of the NCS-1-carrying liposomes with increasing concentrations of the unlabeled protein.

### 2.7. Phospholipid Binding Assay

Binding of NCS-1 to immobilized phospholipids was performed by dot-blot analysis using the PIP Strip™ assay kit (Thermo Fisher, Waltham, MA, USA), according to directions from the literature [8]. The strips were blocked in Tris-buffered saline with 0.1% Tween-20 (TBST) containing 3% fatty acid-free BSA and 1 mM CaCl_2_ for 1 h and then incubated overnight at 4 °C with 0.5 μg/mL NCS-1 in the same buffer. The membranes were washed 3 times with the blocking buffer and the bound protein was stained with anti-NCS-1 antibodies and visualized in the same manner as the other immunoblots.

### 2.8. Western Blotting

Western blotting was performed as described in [42]. Staining of membrane- or phospholipid-bound NCS-1 was performed using rabbit polyclonal (monospecific) antibodies (1:15,000 in TBST) and secondary goat anti-rabbit peroxidase-conjugated antibodies (1:2000 in TBST). The bands were visualized using the enhanced chemiluminescence (ECL) reagent kit and ChemiDoc™ XRS+ gel documentation system (Bio-Rad, Hercules, CA, USA).

### 2.9. Molecular Docking

The structures of ligands were prepared in the Avogadro modeling package [43]. Fatty acid tails were truncated up to ethyl. Receptor structures were subtracted from PDB (2LCP). Both ligands and receptors were preprocessed with AutoDock Tools [43]. Blind flexible docking was performed with QuickVina W, which is optimized for docking in the whole protein [43]. Docking cell of size 80 × 50 × 50 angstroms was used. Each docking run for each of 20 receptor structures produced 20 poses, making 400 in total. The center of mass calculations and visualization were performed with ODDT [44] and PyMOL Molecular Graphics System (Schrödinger, New York, NY, USA).

### 2.10. Statistical Analysis

All data were based on at least three independent experiments and analyzed by the mean standard error (SE) method. Statistical significance was assessed by two-tailed Mann–Whitney U test. The probability of 0.05 was considered significant.

## 3. Results

### 3.1. Binding of NCS-1 to Cellular Membranes

To assess the affinity of NCS-1 to cellular membranes with different lipid composition, we firstly compared its binding to urea-washed hippocampal membranes and membranes of rod outer segments (photoreceptor membranes). These membranes originate from NCS-1-expressing neurons and differ mainly in that in photoreceptor membranes the content of PS is higher than PI, whereas in hippocampal membranes these phospholipids are present in almost equal amounts [15,45]. According to the results of the equilibrium centrifugation assay (Figure 1a), myristoylated NCS-1 binds to photoreceptor and hippocampal membranes with similar affinity. In both cases, the binding seems slightly enhanced in the presence of Ca^2+^, though this effect is statistically insignificant (Figure 1b). Interestingly, unmyristoylated NCS-1 exhibits a higher affinity to photoreceptor membranes than to hippocampal membranes. Consistently, the difference in membrane binding between myristoylated and unmyristoylated protein is reliably more pronounced in the case of hippocampal membranes regardless of the presence of calcium. These data indicate that the interaction of NCS-1 with cellular membranes is indeed sensitive to their phospholipid composition. 

### 3.2. Binding of NCS-1 to Liposomes

We next assessed the affinity of NCS-1 to each of the major membrane phospholipids, using the liposome-binding assay. To this end, we monitored the interaction of fluorescently tagged NCS-1 with multilamellar liposomes prepared from PS, PC, PE or PI. In general, affinity of myristoylated protein to phospholipids decreases in a row PS > PI >> PC > PE (Figure 2a). For instance, the binding of Ca^2+^-saturated NCS-1 to PS is more than 5-fold stronger than to PE. The titration of the myristoylated NCS-1-carrying PI liposomes with increasing concentrations of the unlabeled protein in the presence of Ca^2+^ revealed dissociation constant of 3 μM of the protein (Figure 2b). Notably, the interaction seems to be highly cooperative. Similarly to previous experiments with cellular membranes, myristoylated NCS-1 displays a slight Ca^2+^-dependency of the binding to all phospholipids.

Even though the binding of the unmyristoylated NCS-1 to liposomes is substantially (around 10-fold) weaker than in the case of the myristoylated protein, the overall phospholipid preference of NCS-1 is independent of the presence of the myristoyl group (Figure 2a,c). The only differences are that unmyristoylated NCS-1 exhibits somewhat stronger binding to PS in the presence of calcium and has a lower relative affinity to PI as compared to PS.

### 3.3. Obtaining and Characterization of NCS-1 N-Terminal Mutants

Our results indicate that the interaction of NCS-1 with phospholipid bilayers involves not only the myristoyl group but also certain sites in the protein providing its sensitivity to different membrane content. Since phospholipid-binding amino acids of NCS-1 can be represented by lysine in positions 3, 7 and 9 of the polypeptide chain (see Section 1), we created genetic constructs encoding the protein with single, double or triple substitutions of these residues with glutamic acid. In this case, we faced methodical limitation, namely inability to produce myristoylated forms of the mutants due to substitution of amino acids critical for recognition of NCS-1 by N-myristoyl transferase-1 [46]. Yet, given that NCS-1 is more sensitive to phospholipid composition of the membrane in the absence of myristoyl group (see Figure 1) and that both forms of the protein display similar phospholipid-binding pattern (see Figure 2), only unmyristoylated forms of NCS-1 mutants were prepared (the mutants were expressed in *E. coli* and purified from bacterial lysates in the same fashion as NCS-1^WT^) and used in the following studies.

To explore the possible structural effects of the introduced amino acid substitutions, we compared the thermal stability of the obtained NCS-1 mutants with that of the wild type protein via monitoring temperature dependence of maximum wavelength of their intrinsic fluorescence. It was found that similarly to NCS-1^WT^ the unfolding temperature of all mutants in the presence of Ca^2+^ exceeds 90 °C (data not shown). In the absence of Ca^2+^, most of the NCS-1 variants also display very similar unfolding profiles with mid-transition temperature varying within the 61–65 °C range (Figure 3a). An exception is K3,7E, which exhibits a decrease in mid-transition temperature to 55 °C (Figure 3b). Yet, the significant stabilization of this mutant in the presence of calcium (data not shown) indicates that it still binds Ca^2+^ and the core structure of this protein is largely unaffected by the mutations. We conclude that all the mutants can be used in the further comparative studies, but special emphasis should be given to the effects produced by Ca^2+^-free form of K3,7E.

### 3.4. Binding of NCS-1 N-Terminal Mutants to Cellular Membranes

We next examined how the N-terminal mutations in non-myristoylated NCS-1 would affect its binding to photoreceptor and hippocampal membranes. As can be seen from Figure 4a, the charge inversion in positions 3, 7 and 9 of the polypeptide chain leads to suppression of the NCS-1 binding to photoreceptor membranes in a cumulative manner. Thus, single-point K→E mutants exhibit the same affinity as the wild type protein, whereas the affinity of multiple-point mutants in the presence of Ca^2+^ decreases in a row WT > K3,7E > K3,9E > K7,9E > K3,7,9E (Figure 4a). In particular, the binding of K3,7,9E was reduced two-fold. Regarding hippocampal membranes, the binding of unmyristoylated NCS-1 is generally much weaker and the effects of the mutations are indistinguishable regardless of the presence of Ca^2+^ (Figure 4b). These results agree with our observations that unmyristoylated NCS-1 displays stronger preference to negatively charged phospholipids PS and PI (see Figure 2), which are considerably more abundant in photoreceptor membranes than in hippocampal membranes [15,18]. Based on the data obtained, we can suggest that membrane components enabling capturing of unmyristoylated NCS1 are contained in higher amounts in photoreceptor membranes than in membranes of the hippocampus and these components are likely negatively charged.

### 3.5. Binding of NCS-1 and Its N-Terminal Mutants to Individual Phospholipids

To get further insight into the mechanism of recognition of membrane components by NCS-1, we monitored the binding of its wild type and mutant forms to immobilized monolayers consisting of fourteen different phospholipids in the presence of Ca^2+^ using a dot-blot assay (Figure 5a). In accordance with the results of liposome-binding assay, unmyristoylated NCS-1 exhibits weak interaction with PE and PC but binds to negatively charged PS and, to lesser extent, phosphatidic acid (PA) (Figure 5b). Yet, the protein displays a very low affinity to immobilized PI, which contrasts to its interaction with PI-containing liposomes, apparently reflecting differences in NCS-1 binding to monolayers and bilayers consisting of this phospholipid. Notably, the substitution of the N-terminal lysines in NCS-1 reliably reduces its binding to immobilized PS (Figure 5b) thereby confirming the importance of these residues in recognition of negatively-charged phospholipids.

Remarkably, NCS-1 demonstrates pronounced binding to phosphoinositides, especially to PI3P, which demonstrates 8-fold greater affinity to the protein than PS (Figure 5b). The binding of NCS-1 to these signaling phospholipids decreases in the following order: PI3P >> PI(3,5)P_2_ ≈ PI(4,5)P_2_ ≈ PI(3,4,5)P_3_ ≈ PI4P ≈ PI5P >> PI(3,4)P_2_. Unexpectedly, the introduction of N-terminal charge-reversing mutations in NCS-1 produces only moderate effects on its interaction with phosphoinositides, representing mainly a decrease of selectivity of their recognition (Figure 5c). Indeed, all mutants demonstrate similar binding to PI3P (Figure 5c, inset), but somewhat lower relative preference to this phospholipid over PI4P and PI5P, as well as visibly enhanced ability to bind PI(3,5)P_2_. Thus, NCS-1 is capable of highly specific recognition of phosphoinositides, especially PI3P. Meanwhile, K3, K7, and K9 are unlikely to solely ensure this recognition but could help to maintain the structure of the highly selective PI3P-binding site.

Given that NCS-1 is myristoylated in the cell, we next verified if the native form of the protein displays the same profile of phospholipid recognition as its unmyristoylated variant. As can be seen from Figure 5a, the general phospholipid-binding pattern of unmyristoylated wild type and mutant NCS-1 including its high preference to PI3P is retained by the myristoylated form of the protein, which additionally demonstrates more pronounced affinity to PI5P and PA.

### 3.6. Prediction of PI3P-Binding Sites in Ca^2+^-Loaded NCS-1

To elucidate possible phosphoinositide-binding site(s) we performed global blind flexible docking of phosphoinositides into the ensemble of unmyristoylated NCS-1 NMR structures (PDB: 2LCP) (Figure 6a). Since NCS-1 demonstrated the highest affinity to PI3P and the most pronounced selectivity between phosphatidylinositol-monophosphates, the docking ligands were limited to PI3P, PI4P, and PI5P. Given that specific recognition of the phosphoinositides by the protein might involve only their polar groups [47], the fatty acid tails were truncated up to ethyl. All three ligands expectedly demonstrate non-specific interaction with the open hydrophobic pocket of the Ca^2+^-bound NCS-1 (see Section 1). Meanwhile, the distribution of centers of mass of the analyzed phosphoinositides indicates the presence of a site exhibiting highly specificity towards PI3P, but not PI4P and PI5P (Figure 6b). This site is unique for the 10th frame of the PDB entry and is located in the proximity of the N-terminus, involving several charged residues, specifically S4, K7, Q28, Q29 and K32 (Figure 6b). Among these, K7 and K32 make electrostatic contacts with the phosphate group in the 3rd position of the inositol ring, while S4, Q28, and Q29 form hydrogen bonds with the phosphate and the inositol moieties. Interestingly, K7 and K32 are conserved within the NCS family, whereas the combination of S4, Q29, and Q30 is unique for NCS-1 and its yeast homologs frequenins (Figure 6c), suggesting that PI3P binding represents a specific feature of frequenin branch of the NCS family.

In conclusion, our data suggest that the N-terminus of NCS-1 supports both general membrane anchoring of the protein by providing myristoyl group and positively charged residues and its highly specific binding to PI3P apparently by forming the respective site. The second phospholipid-binding mode is a novel feature of NCS-1, which may allow targeting the protein to particular phospholipids and ensure its involvement in phosphoinositide-regulated signaling pathways.

## 4. Discussion

In this study we compared the ability of NCS-1 to interact with membranes of hippocampal and photoreceptor neurons, which are known to express this protein [22,23,48]. Under in vitro conditions, myristoylated NCS-1 does not discriminate between these membranes and their binding is Ca^2+^-independent, in agreement with a number of previous reports [24,27,28]. Although in both cases the interaction is enhanced by myristoylation of the protein as was previously demonstrated in cellular models [10], such modification is more critical for NCS-1 binding to hippocampal membranes. This difference could be related to the distinct phospholipid composition of photoreceptor and hippocampal membranes. Thus, both of them contain PC and PE in equal proportions (around 40% of total phospholipid content), but negatively charged PS and PI (including its products) together account for ~16%–25% of total phospholipid content of photoreceptor membranes and only for ~5%–10% of total phospholipid content of hippocampal membranes [15,18]. Thus, unmyristoylated NCS-1 can be attracted to the negatively charged surface of photoreceptor membranes by some positively charged residues of the protein, while in hippocampal membranes this electrostatic attraction will be considerably weaker.

Indeed, our liposome binding experiments demonstrate that NCS-1 has a higher affinity to PS and PI than to PE and PC. The dissociation constant for the interaction of myristoylated NCS-1 with PI is 3 μM, which agrees with estimations of total NCS-1 concentration in brain and retina (0.5–5 μM) [49]. The preference of the protein to negatively charged phospholipids is preserved in its unmyristoylated form, even though the overall efficiency of liposome binding is decreased. Interestingly, phospholipid binding appears to be more myristoyl-dependent in case of PI, as compared to PS. The binding of NCS-1 to PI is elevated 20-fold upon myristoylation, while only ~10-fold in case of PS. This could be attributed to the differences in size and charge localization between the head groups of these phospholipids. As was shown for recoverin [13], myristoylation may cause NCS-1 to assume certain position on the membrane surface and this exact orientation could more prominently enhance its binding of PI, as compared to PS. It should be noted that these differences in behavior of myristoylated and unmyristoylated NCS-1 might not essentially affect the results of native membrane binding experiments (including comparison of the membrane affinities of the NCS-1 mutants, see below), because there is considerably less PI than PS in both photoreceptor and hippocampal membranes [15,18]. Nevertheless, in general, our data indicate that the contribution of PS and PI to overall attraction of NCS-1 to neuronal membranes would be significant, making the protein highly sensitive to the abundance of these phospholipids. Indeed, in living cells, NCS-1 was shown to predominantly localize on the inner surface of the plasma membrane, which is enriched in PS compared to other membrane types [50]. PS commonly serves as an attractor for the signaling proteins, facilitating the co-localization of protein and lipid components of various signaling pathways (for review, see [51]).

The residues that potentially determine NCS-1 affinity to negatively charged phospholipids can be located in the N-terminal region of the protein in the proximity of its myristoyl group and may include K3, K7, and K9 as the respective residues were shown to contribute to membrane localization of the other NCS proteins, recoverin and hippocalcin [13,19]. We confirmed this suggestion by using NCS-1 mutants, where N-terminal positively charged lysine residues were substituted by glutamic acid. According to thermal stability tests based on monitoring of temperature dependences of intrinsic (tryptophan) fluorescence of the protein [24], these substitutions do not affect the overall structure of NCS-1. Meanwhile, their stepwise introduction expectedly downregulates the binding of NCS-1 to photoreceptor membranes in cumulative manner. While single K→E mutations do not significantly affect the affinity of the protein to these membranes, double mutations have a notable negative effect on the binding, whereas the triple mutation inhibits it more than 2-fold. By contrast, hippocampal membranes initially demonstrate a very low rate of NCS-1 association, which is hardly affected by these mutations. The latter could be related to the fact that, based on our liposomes binding data, PS would account for ~50% of unmyristoylated NCS-1 binding to photoreceptor membranes but only for ~20% in case of hippocampal membranes. As a result, the latter interaction would not only be significantly weaker but also less affected by the charge inversions, as observed in our experiments. These findings point to the importance of the N-terminal region of NCS-1 in recognition of negatively charged cellular membranes, associated with the enhanced affinity of NCS-1 to certain phospholipids. Consistently, in our dot-blot experiments the binding of NCS-1 to PS is indeed decreased upon reducing the positive net charge of the N-terminal region of NCS-1 by mutation of its K3, K7, and K9. While NCS-1 lacks a conventional β-sheet PS-binding domain (such as C2 domain), there is growing evidence for the role of comparatively simple structural motifs containing conserved basic residues in PS detection and binding [52,53,54].

Generally, proteins do not display a high specific affinity towards PS. However, for some proteins, membrane context enriched in negatively charged phospholipids can facilitate specific recognition of low abundance signaling phospholipids, namely phosphoinositides [55]. Remarkably, unmyristoylated NCS-1 exhibits a pronounced affinity to phosphoinositides, specifically PI3P (~8-fold higher affinity to PI3P as compared to PS). Furthermore, this highly specific interaction is similarly peculiar to native-like myristoylated protein. Previous studies using a living cells model reported preferential binding of myristoylated NCS-1 to PI(4,5)P_2_ over PI4P [8]. Although we observed the same tendency, the interaction of NCS-1 with these phospholipids is 4-40 times lower than with PI3P, which clearly points to the latter as the main binding partner of the protein in cellular membranes. Interestingly, another NCS protein, hippocalcin, demonstrates pronounced binding to phosphoinositides without selectivity to any of them in vitro and exhibits a higher affinity to PI(4,5)P_2_ over PI4P in a cellular model. While these data are somewhat contradictory, they clearly reflect differences in the organization of phospholipid-binding sites of hippocalcin and NCS-1 [8]. Consistently, hippocalcin was previously shown to bind phosphoinositides via short myristoylated N-terminal peptide [8], whereas our current observations suggest that NCS-1 possess a phospholipid-binding site outside of the immediate proximity of the myristoyl group, as the substitution of the N-terminal lysine residues does not affect the general pattern of phospholipid binding by NCS-1.

This suggestion is supported by the results of our molecular docking aimed at predicting phosphoinositide-binding site(s) within the NCS-1 structure. Although most of the phosphoinositides dock non-specifically into the exposed hydrophobic pocket of Ca^2+^-bound NCS-1, PI3P demonstrates specific binding to a site involving a loop preceding first α-helix of the protein as well as its second α-helix. This site is comprised of five residues, among which K7 and K32 form a positively charged pocket harboring phosphate group, whereas S4, Q28, and Q29 interact with both phosphate and inositol moieties of PI3P through a network of hydrogen bonds. The site is located close to the N-terminus of NCS-1 and contains one of its N-terminal lysine residues (K7), which could explain moderate alterations in phosphoinositide binding properties of charge-inverting mutants of the protein, examined in our study. However, three of the five residues within the site (Q28, Q29, and K32) belong to a more rigid structure involving the first α-helix of the first EF-hand motif of NCS-1. Thus, it seems possible that NCS-1 exhibits two modes of membrane interaction. The first mode is myristoyl-dependent binding to cellular membranes, which involves N-terminal lysine residues ensuring preference of the protein to membranes enriched in acidic phospholipids. A similar mode of interaction was previously suggested for another NCS protein, recoverin, the binding of which was shown to involve both myristoyl group and N-terminal positively charged residues (in recoverin, these are K37, R43 and K84) providing specific spatial orientation of the protein on the membrane [13]. The second mode of NCS-1 interaction with membranes includes its highly specific ‘signaling’ binding to phosphoinositides, especially PI3P, and this binding likely involves the above-described site in NCS-1 structure. The residues forming this site are generally variable among the NCS proteins, which may underlie their different affinities to signaling phosphoinositides and other phospholipids. For instance, hippocalcin contains the same residues as K7, Q28, and K32 of NCS-1, but S4 and Q29 are unique for the latter, which could explain its preference of different phosphoinositides despite similar subcellular localization with hippocalcin [8,19].

Notably, the overall site is conserved in proteins of frequenin branch of the NCS family, including NCS-1 and yeast frequenins, which were first shown to participate in the regulation of phosphoinositide metabolism via regulation of phosphatidyl-inositol-4-kinase-1 [56,57]. The ability of NCS-1 to distinguish isolated phosphoinositides with high precision could be explained by the fact that NCS-1 has evolved as a ubiquitous Ca^2+^-sensor with a wide array of signaling partners. This flexibility, in addition to extremely high Ca^2+^-sensitivity and constitutive membrane localization, calls for additional mechanisms of regulation to avoid indiscriminate binding of target proteins by NCS-1. High affinity to phosphoinositides could help compartmentalize NCS-1 with its targets in the appropriate conditions, as phosphoinositides are distributed in distinct spatial patterns, which fluctuate in response to stimuli (reviewed in [58]). For example, the ability to discern PI3P and PI4P could be of great importance for NCS-1, as by binding PI4P it could sequester the substrate of its own regulatory target phosphatidylinositol-4-kinase βIII, that NCS-1 is supposed to activate [59]. By contrast, the other NCS proteins could possess different or less pronounced specificity to phosphoinositides. For instance, recoverin and hippocalcin represent more differentiated NCSs with a Ca^2+^-myristoyl switch, which could minimize the chances of their engaging in non-specific regulation. Consistently, there is no data reporting the preference of recoverin to phosphoinositides, whereas hippocalcin displays a more lenient preference of these signaling phospholipids as compared to NCS-1. Thus, hippocalcin readily associates with major phosphoinositide PI(4,5)P_2_, which accounts for as much as 1% of total plasma membrane lipids and is 25-fold more abundant than PI3P [60].

Phosphoinositides represent one of the most universal types of signaling molecules in neurons. The function of major phosphoinositides PI4P and PI(4,5)P_2_ includes, but is not limited to, providing the substrate for generation of second messengers inositol-triphosphate and diacylglycerol [61]. In turn, all PI-3-phosphates (including PI3P) and PI5P are classified as low abundance phosphoinositides that can recruit proteins to a specific location on the membrane surface, organizing them into signaling domains [1]. In addition, both major and low abundance phosphoinositides can be sequestered by specific regulatory proteins to modulate their availability to signaling targets [62,63]. PI3P has the highest number of confirmed specific binding partners, in contrast to the more abundant phosphoinositides, especially PI(4,5)P_2_, which interacts less specifically with almost any phosphoinositide-binding protein [47]. This fact points to the potential physiological significance of the observed specific interaction of NCS-1 with PI3P, which, according to our data, is 4-fold and 40-fold stronger than the binding of the protein to more abundant PI(4,5)P_2_ and PI4P, respectively.

Indeed, there is a notable overlap in localization and function between NCS-1 and PI3P. NCS-1 is well-recognized as a multifunctional regulator of secretion (reviewed in [64]). It predominantly localizes on trans-Golgi network and plasma membranes and participates in the regulation of synaptic vesicle release, receptor internalization and phosphoinositide biosynthesis [59,65]. PI3P is also localized in Golgi and synapses, where it is involved in the regulation of endosomal sorting and trafficking [66,67]. Furthermore, it is known to regulate the surface levels and clusterization of neurotransmitter receptors in hippocampal neurons [68]. Recently, the NCS-1 expression level was found to be dysregulated in patients with Alzheimer’s disease [69]. Consistently, PI3P deficiency is associated with altered amyloid precursor processing observed in Alzheimer’s disease [70]. In addition, PI3P was recognized as a regulator of autophagy, thereby playing a key role in mechanisms of neuronal survival [71]. Interestingly, both NCS-1 and PI3P were found in outer segments of photoreceptor neurons, where the function of NCS-1 remains unclear. Yet, the activity of PI3-kinase responsible for conversion of PI to PI3P was detected in photoreceptor outer segments and was found to be activated by light, while PI3P itself was found to be involved in the vesicular membrane targeting of visual receptor rhodopsin [72,73,74]. These data suggest that NCS-1 can bind to PI3P in membrane-rich photoreceptors and this binding may be of physiological significance. It added that in our experiments myristoylated NCS-1 exhibits relatively strong binding to another low abundance phosphoinositide, PI5P. This phospholipid was shown to be involved in endosomal trafficking [75], but its general function in neurons is yet to be fully understood.

It cannot be ruled out that the revealed high-affinity binding of NCS-1 to phosphatidylinositol phosphates ensures its specific membrane localization rather than directly contributes to phosphoinositide-dependent signaling. For instance, in living cells, NCS-1, similarly to hippocalcin [8], may be predominantly associated with different phosphatidylinositol phosphates (including PI3P, PI5P, etc.), which may govern its targeting to certain sites on plasma or Golgi membranes. Such localization can, in turn, compartmentalize NCS-1 with its binding partners, thereby maintaining Ca^2+^-dependent regulatory function of the protein.

Future studies are needed to confirm the ability of NCS-1 to specifically bind PI3P and the other phosphoinositides in a more competitive medium in vivo and to reveal exact physiological outcomes of this interaction. Thus, PI3P-specific fluorescent biosensors (reviewed in [76] and exemplified with PI(4,5)P_2_- and PI4P-specific biosensors in [8]) could be applied to detect the formation of NCS-1/PI3P complex in cells. Although our molecular modeling experiments convincingly predict PI3P-binding site in NCS-1, the structure of this site and the mechanism underlying specific recognition of certain phosphatidylinositol phosphates by the protein require further verification, using the respective NCS-1 mutants.

## 5. Conclusions

In this study, we have demonstrated that ubiquitous neuronal signaling protein NCS-1 exhibits two modes of membrane interaction. The first mode represents myristoyl-dependent binding of NCS-1 to cellular membranes (with low micromolar affinity), which involves N-terminal lysine residues K3, K7, and/or K9 ensuring preference of the protein to membranes enriched in acidic phospholipids. The second mode includes its highly specific ‘signaling’ binding to phosphoinositides (apparently with submicromolar affinity), especially PI3P, and this binding is generally myristoyl-independent and likely involves a novel specific site located in N-terminal region of the protein. Both variants of interaction are mediated by the N-terminal region of NCS-1, which from the one side ensures the general membrane binding of the protein by providing myristoyl group and positively charged residues, while from the other side may support highly specific binding to PI3P by forming the respective site. This potential site is conserved in proteins of the frequenin branch of the NCS family, suggesting that the ability to recognize PI3P is a specific feature of these particular NCS proteins, which are known to participate in the regulation of phosphoinositide metabolism via regulation of phosphatidyl-inositol-4-kinase-1. A notable overlap in localization and function between NCS-1 and PI3P suggests that their interaction can be of physiological importance. Thus, both biomolecules are involved in vesicle sorting and trafficking, co-localize in Golgi membranes and synapses of neurons, and becomes dysregulated in neurodegenerative disease. Given the interactions revealed in this study, one could suggest the possible involvement of NCS-1 and its phosphoinositide target(s) in the same signaling pathways governing neuronal function and survival.

## Figures and Tables

**Figure 1 biomolecules-10-00164-f001:**
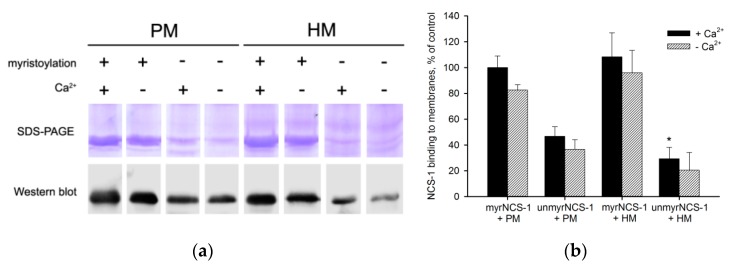
Binding of myristoylated and unmyristoylated neuronal calcium sensor-1 (NCS-1) to urea-washed photoreceptor membranes (PM) and hippocampal membranes (HM) in the presence of 2 mM CaCl_2_ (“+Ca^2+^”) or 2 mM EGTA (“−Ca^2+^”). (**a**) Representative results of SDS-PAGE and Western blotting of membrane-bound NCS-1. (**b**) Relative efficiency of membrane binding. The fraction of myristoylated NCS-1 bound to PM in the presence of Ca^2+^ was taken as 100%. * *p* < 0.05 as compared to binding of unmyristoylated NCS-1 to PM in the presence of Ca^2+^.

**Figure 2 biomolecules-10-00164-f002:**
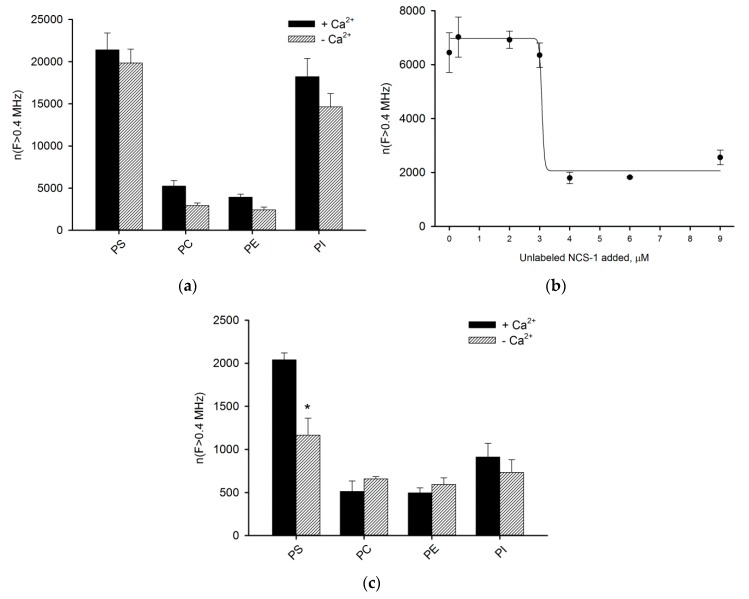
Binding of sulfo-cyanine-3-NCS-1 to multilamellar liposomes prepared from major cellular phospholipids, assessed by fluorescence correlation spectroscopy (FCS). (**a**) Binding efficiency of myristoylated NCS-1 to liposomes is represented by the total count of high-intensity FCS peaks exceeding 0.4 MHz threshold. (**b**) Titration of phosphatidylinositol (PI) liposomes carrying Ca^2+^-bound myristoylated NCS-1 with increasing concentrations of unlabeled NCS-1. (**c**) Binding efficiency of unmyristoylated NCS-1 to liposomes. * *p* < 0.05 as compared to binding of unmyristoylated NCS-1 to phosphatidylserine (PS) in the presence of Ca^2+^.

**Figure 3 biomolecules-10-00164-f003:**
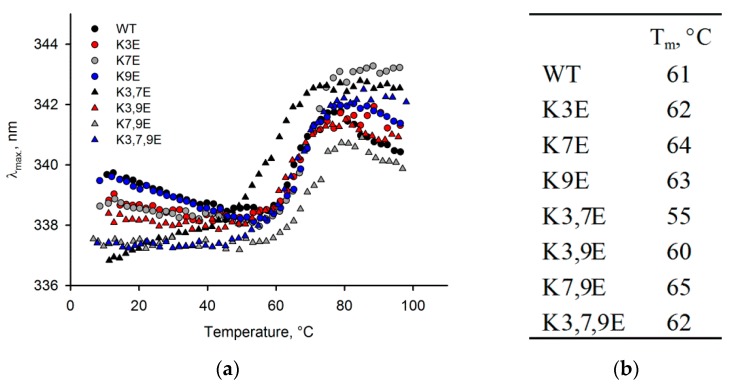
Thermal unfolding of unmyristoylated Ca^2+^-free Mg^2+^-bound NCS-1 and its mutants. (**a**) Temperature dependence of maximum wavelength of tryptophan fluorescence (λ_max_); (**b**) Mid-transition temperatures (T_m_) of the protein unfolding.

**Figure 4 biomolecules-10-00164-f004:**
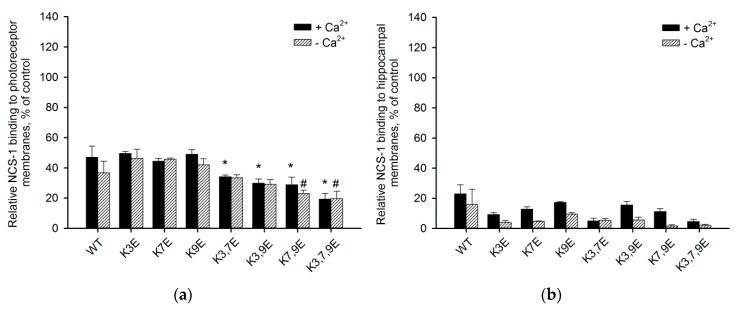
The efficiency of binding of unmyristoylated NCS-1 N-terminal mutants to (**a**) photoreceptor and (**b**) hippocampal membranes. The fraction of myristoylated NCS-1 bound to PM in the presence of Ca^2+^ was taken as 100%. * *p* < 0.05, as compared to binding of NCS-1^WT^ in the presence of Ca^2+^; #—in the absence of Ca^2+^.

**Figure 5 biomolecules-10-00164-f005:**
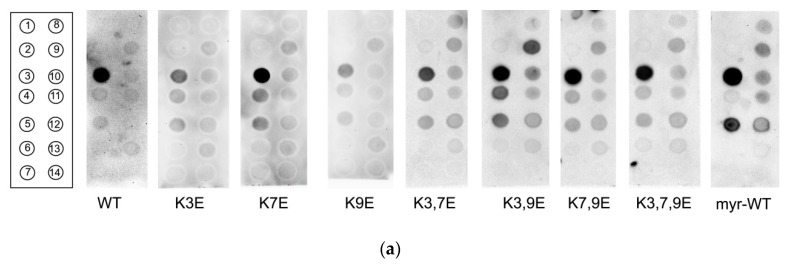
Binding of unmyristoylated Ca^2+^-bound NCS-1^WT^, its N-terminal mutants, and myristoylated NCS-1^WT^ to immobilized phospholipids determined using PIP-Strip™ assay. (**a**) Immunostained dot-blot membranes. Positions of lipid species are indicated in the left panel: 1—lysophospatidylcholine; 2—phosphatidylinositol (PI); 3—phosphatidylinositol-3-phosphate (PI3P); 4—phosphatidylinositol-4-phosphate; 5—phosphatidylinositol-5-phosphate; 6—phosphatidylethanolamine (PE); 7—phosphatidylcholine (PC); 8—phosphatidylinositol-3,4-bisphosphate; 9—phosphatidylinositol-3,5-bisphosphate; 10—phosphatidylinositol-4,5-bisphosphate; 11—phosphatidylinositol-3,4,5-triphosphate; 12—phosphatidic acid (PA); 13—phosphatidylserine (PS); 14—blank. (**b**) The efficiency of binding of unmyristoylated NCS-1^WT^ and K3,7,9E mutant to major phospholipids of cellular membranes. * *p* < 0.05, as compared to binding of NCS-1^WT^ to PS. (**c**) The efficiency of binding of unmyristoylated NCS-1^WT^ and its mutants to phosphoinositides. The fraction of unmyristoylated NCS-1 bound to PI3P in the presence of Ca^2+^ was taken as 100% (relative efficiency of PI3P binding by the mutants is shown in inset).

**Figure 6 biomolecules-10-00164-f006:**
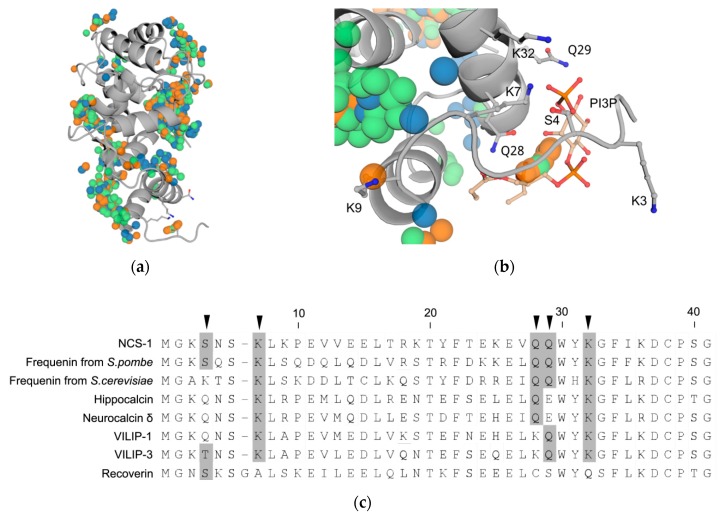
Molecular docking of phosphatidylinositol-monophosphates into unmyristoylated NCS-1 (PDB: 2LCP; frame 10). (**a**) Centers of mass of PI3P (orange spheres), PI4P (green spheres) and PI5P (blue spheres) docked into NCS-1. (**b**) Putative PI3P-binding site. NCS-1 residues making electrostatic contacts or hydrogen bonds with PI3P as well as conserved lysine residues K3 and K9 are shown as sticks. (**c**) Sequence alignment of human NCS-1, its yeast homologs frequenins, and closely related members of the NCS family. Residues of the putative PI3P-binding site are marked with black arrowheads, conserved residues in these positions are highlighted in gray.

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
