# Peer review of "Membrane Binding of Neuronal Calcium Sensor-1: Highly Specific Interaction with Phosphatidylinositol-3-Phosphate"

_biomolecules, 2020, doi:10.3390/biom10020164_

Round 1

Reviewer 1 Report

The manuscript (658853) presents in vitro binding that probes the interaction of neuronal calcium sensor-1 (NCS-1) with natural membranes of different lipid composition as well as individual phospholipids and the phospholipid preference of the protein.  The data suggest that NCS-1 binds to neuronal membranes in a Ca2+-independent manner, and NCS-1 binding to photoreceptor membranes is less dependent on myristoylation, but instead the binding is attenuated by the replacement of K3, K7, and/or K9 of the protein by glutamic acid.  The results suggest that NCS-1 preferentially binds to negatively charged phospholipids; however, the experimental data are not quantitative enough and are somewhat incomplete. The in vitro membrane binding assays (Figs. 1, 3 and 4) are incapable of quantitatively measuring the binding dissociation constant (Kd).  The paper would be dramatically improved if the qualitative binding assays in Figs. 1, 3 and 4 were replaced by a more quantitative binding assay like surface plasmon resonance (SPR) performed on lipid bilayer membranes as described by Dell’Orco (Anal Chem. 2012 Mar 20;84(6):2982-9. doi: 10.1021/ac300213j. Epub 2012 Mar 8). For example, what is the Kd of NCS-1 binding to the various lipid bilayers and does this Kd occur in the physiological concentration range of NCS-1 in neurons?  If the Kd value is greater than 100 micromolar, then such low affinity would suggest a non-specific electrostatic interaction that is most likely not physiologically relevant. However, is the Kd is in the sub-micromolar range, then this higher affinity binding would be consistent with a physiological interaction. Also, the paper fails to demonstrate whether the predicted protein-lipid interactions from this study are actually occurring in vivo.  Do the NCS-1 mutants from the current study (K3,7,9E) have any effect on the physiological function of NCS-1?  There needs to be some demonstration in this paper that the predicted protein-lipid interactions are physiologically relevant and the binding affinity is occurring in the physiological range.  What is the evidence that these interactions serve a physiological role?  The docking calculation in Fig. 5 suggests that Q28, Q29 and K32 are making direct contacts with PI-3P.  This structural interaction needs to be experimentally verified.  For example, it should be shown in this paper whether the NCS-1 mutants (Q28A, Q29A and K32E) have an effect on NCS-1 binding to PI-3P. 

Author Response

The manuscript (658853) presents in vitro binding that probes the interaction of neuronal calcium sensor-1 (NCS-1) with natural membranes of different lipid composition as well as individual phospholipids and the phospholipid preference of the protein. The data suggest that NCS-1 binds to neuronal membranes in a Ca2+-independent manner, and NCS-1 binding to photoreceptor membranes is less dependent on myristoylation, but instead the binding is attenuated by the replacement of K3, K7, and/or K9 of the protein by glutamic acid.

ANSWER:

We truly appreciate the detailed analysis of the study. Please, find our response below. For the convenience, the introduced changes are highlighted in the updated manuscript with blue font.

The results suggest that NCS-1 preferentially binds to negatively charged phospholipids; however, the experimental data are not quantitative enough and are somewhat incomplete. The in vitro membrane binding assays (Figs. 1, 3 and 4) are incapable of quantitatively measuring the binding dissociation constant (Kd).  The paper would be dramatically improved if the qualitative binding assays in Figs. 1, 3 and 4 were replaced by a more quantitative binding assay like surface plasmon resonance (SPR) performed on lipid bilayer membranes as described by Dell’Orco (Anal Chem. 2012 Mar 20;84(6):2982-9. doi: 10.1021/ac300213j. Epub 2012 Mar 8). For example, what is the Kd of NCS-1 binding to the various lipid bilayers and does this Kd occur in the physiological concentration range of NCS-1 in neurons?  If the Kd value is greater than 100 micromolar, then such low affinity would suggest a non-specific electrostatic interaction that is most likely not physiologically relevant. However, is the Kd is in the sub-micromolar range, then this higher affinity binding would be consistent with a physiological interaction.

ANSWER:

We thank the reviewer for this recommendation. To probe NCS-1 affinity to phospholipids in a more quantitative way, we performed a number of additional experiments (see Results section 3.2 and Methods section 2.6 in the updated manuscript), using NCS-1 with fluorescent tag on its cysteine-38 and multilamellar liposomes made from major phospholipids: phosphatidylserine, phosphatidylethanolamine, phosphatidylcholine and phosphatidylinositol. The binding of myristoylated and unmyristoylated NCS-1 to the liposomes under different conditions (+/- Ca2+) was monitored using fluorescence correlation spectroscopy (FCS) method as described by Antonenko et al. [PMID:27837242]. The results are represented in the new Figure 2a-c. According to these data, NCS-1 indeed demonstrates strong, predominantly Ca2+-independent preference to negatively charged phospholipids PS and PI over more abundant but zwitterionic/neutral PE and PC. The same pattern is retained for the unmyristoylated form of the protein, too, even though the overall intensity of the binding is lower, which additionally justifies using unmyristoylated forms of NCS-1 and its mutants in our subsequent experiments. Furthermore, we managed to measure Kd value characterizing liposome binding of myristoylated NCS-1 by titrating the liposomes with increasing concentrations of unlabeled protein and monitoring the decline of the signal. Kd for NCS-1/PI interaction was estimated at 3 μM, which is within the range of the protein concentration in neurons (up to 5 μM; [PMID 8799187]). Overall, it can be concluded that NCS-1 in low micromolar concentrations effectively binds negatively charged phospholipids (as was correctly presumed from membrane binding studies) and the binding is therefore consistent with a physiological interaction.

Also, the paper fails to demonstrate whether the predicted protein-lipid interactions from this study are actually occurring in vivo.  Do the NCS-1 mutants from the current study (K3,7,9E) have any effect on the physiological function of NCS-1?  There needs to be some demonstration in this paper that the predicted protein-lipid interactions are physiologically relevant and the binding affinity is occurring in the physiological range. What is the evidence that these interactions serve a physiological role?

ANSWER:

We want to thank the reviewer for this important question. There are a number of studies demonstrating that NCS-1 interaction with phospholipid membranes occurs in vivo [PMID: 15973684, 12127101, 10022960, 9712909, 11006299]. Furthermore, multiple works report that efficacy and specificity of membrane binding is crucial for the functioning of all NCS proteins including NCS-1 (please see Introduction section). For NCS-1, it was demonstrated that single-point mutation R102Q associated with autistic spectrum disorder distorts membrane interaction of the protein without directly affecting its Ca2+ binding or target recognition [PMID:20479890]. Moreover, recent studies revealed an alternative splice form of NCS-1 lacking both the myristoylation motif and the adjacent N-terminal amino acids [PMID:    27575489]. Our results suggest that this form could have a completely different subcellular localization and/or function, as compared to the major isoform of NCS-1. Overall, our study supports previous indications that NCS1 interacts with lipid membranes and confirms that this interaction is physiologically relevant as the affinity of the protein to the lipids is within the range of its concentration in the cell (please see Figure 2). The main aim of our study was determining the structural basis underlying general membrane binding of NCS-1 and its selectivity towards major phospholipids and phosphoinositides. We believe that we reached this aim as our results provide important new insight into fundamental mechanisms of regulatory specificity of NCS-1 and other NCS proteins, as well as completely new data pointing on highly specific interaction of NCS-1 with crucial signaling phospholipid PI3P and other phosphoinositides. 

Although we agree with the reviewer that our conclusions generally require physiological confirmations, it seems unlikely that employment of our NCS-1 mutants in functional assays would provide much information as the mutations do not affect neither Ca2+-binding nor target recognition sites of the protein [PMID: 22363261]. More likely, the mutations could indirectly affect NCS-1 functionality by disrupting its colocalization on membranes with target proteins. Yet, investigation of this issue would require a cell model with phospholipid content resembling photoreceptor and hippocampal membranes, which might overexpress NCS-1 and each of its mutants together with one of multiple (more than twenty) targets of the protein. We believe that this investigation could be a subject of a separate study.

The docking calculation in Fig. 5 suggests that Q28, Q29 and K32 are making direct contacts with PI-3P. This structural interaction needs to be experimentally verified.  For example, it should be shown in this paper whether the NCS-1 mutants (Q28A, Q29A and K32E) have an effect on NCS-1 binding to PI-3P. 

ANSWER:

We totally agree with the reviewer that the interaction of PI3P with the described specific N-terminal site in NCS-1 needs to be verified in direct experiments. In fact, in order to reliably confirm such complex multipoint (structural) site one might use triple/quadruple mutants as single substitutions may not significantly affect the observed high-affinity interaction. Consistently, we did not found any significant differences in PI3P binding to K7E mutant lucking one of the residues specifically interacting with the phosphoinositide head in silico. On the other hand, in contrast to N-terminal lysines addressed in this study, the residues Q28, Q29 and K32 are located in the protein core and their substitution (especially yielding triple or quadruple mutants with K7 substitution) may indirectly influence PI3P binding by affecting overall structure of the protein. Thus, the rigorous proof of the site requires structural and functional characterization of multiple mutants, which might take more time (Biomolecules provides only 10 days for major revision) and, to our opinion, also deserves a separate study.

Reviewer 2 Report

Review

This study addresses the mechanism of interaction of NCS-1 to membranes and demonstrates different mode of interaction including electrostatic-mediated binding.

The study is well performed overall. However, there are a few issues that should therefore be addressed.

The manuscript is overall well written. However definite article are sometimes missing throughout the manuscript. This needs to be checked.

Line 104: indicate lysine in “the residues”

Lines 235-6. The following sentence is unclear: “Given that the effect of phospholipid content on membrane binding of NCS-1 is better seen in the absence of myristoyl group”. Please rephrase and clarify.

Section 3.4. The authors claim that unmyristoylated NCS-1 binds to PA but this is not evident from the lipid blot (a) and the quantification (b) shows an overall low level. The statement should at least be toned-down in relation to the interaction signal given by PS.

In addition, the importance given to PS should also be toned down. The evidence is not strong in particular from the lipid blot (a), showing the comparison of the WT to the triple mutant. In addition, the graph appears to show a greater decrease with the triple mutant in the graph compared to the blot. This needs to be addressed.

Figure 4. please show a separate quantification graph (from c) for PI3P for the WT and different mutants.

Figure 5: in line with the results suggested by the docking experiment, it would be useful to mutate K32 with or without mutating K7 and test these mutants on lipid blots. K3 and K9 could also be tested.

Figure 4: K3 and K9 have the best effect on PI3P binding and the side chain should be indicated in the 3D structure to show their orientation in relation to the proposed orientation of the lipid, as in Figure 5b. Could these two lysines promote interaction, perhaps in combination with K32?

Discussion. The known mechanism of interaction of PS to other proteins should be discussed as well.

Author Response

The submitted manuscript entitled “Membrane binding of Neuronal Calcium Sensor-1: highly specific interaction with phosphatidylinositol-3-phosphate” by Viktoriia Baksheeva , Ekaterina Nemashkalova , Arthur Zalevsky , Vasily Vladimirov , Natalia Tikhomirova , Pavel Philippov , Andrey Zamyatnin Jr. , Dmitry Zinchenko , Sergey Permyakov and Evgeni Zernii addresses potential membrane interactions of the calcium sensor NCS-1. The authors have utilised a number of N-terminal mutants of the sensor to try to define membrane-binding mechanisms, which Is coupled with an in vitro approach to demonstrate lipid interactions. Overall the manuscript is well presented, clearly written, and the experimental research is clearly defined. However, in my opinion, although the evidence of phosphatidylinositol monophosphate binding is a novel and well supported observation, the work is at a very preliminary stage and would not warrant publication in this journal without significant further supportive evidence.

A number of major concerns are raised which would have to be significantly addressed before consideration for publication, specifically;

ANSWER:

We thank the reviewer for his/her insightful comments and recommendations. Please find our detailed responses below. For the convenience, the introduced changes are highlighted in the updated manuscript with blue font.

The evidence for binding mechanisms presented is based solely on in vitro observations using recombinant protein. These observations also rely on crude membrane preparations from neuronal sources to monitor bound fractions, and also adsorbed lipid binding assays to suggest binding specificity. As the authors discuss the relative lipid compositions of the total cellular membrane preparations at length, and suggest differences in these relate to the protein binding modality, this does not account for differences in lipid compartmentalization and enrichment in the intact cells. Where a crude membrane preparation has a lower lipid composition, this may not be relevant if the composition of a targeted internal lipid compartment is enriched to a significant extent. Analysis within a cell-based context may add supporting evidence, even an over-expressed system utilizing the mutant forms the authors have. Furthermore the use of artificial membranes or liposome/micelle-bound fractions would give better evidence of association.

ANSWER:

As it was suggested by the reviewer, we performed additional experiments (see Results section 3.2 and Methods section 2.6 in the updated manuscript) aimed at characterizing NCS-1 interaction with each of the major phospholipids and providing better evidence that these interactions occur in the physiological range of the protein concentration. We used NCS-1 with fluorescent tag on its cysteine-38 and multilamellar liposomes made from phosphatidylserine, phosphatidylethanolamine, phosphatidylcholine and phosphatidylinositol. The binding of myristoylated and unmyristoylated NCS-1 to the liposomes under different conditions (+/- Ca2+) was monitored using fluorescence correlation spectroscopy (FCS) method as described by Antonenko et al. [PMID 27837242]. The results are presented in the new Figure 2a-c. According to these data, NCS-1 indeed demonstrates strong, predominantly Ca2+-independent interaction with preference to negatively charged phospholipids PS and PI over more abundant but zwitterionic/neutral PE and PC. The same pattern is retained for the unmyristoylated form of the protein, even though the overall intensity of binding is lower. These observations additionally justify using unmyristoylated forms of NCS-1 and its mutants in our subsequent experiments. Furthermore, we managed to estimate Kd of NCS-1/PI complex at 3 μM, which is within range of the protein concentration in neurons [PMID 8799187]. Overall, it can be concluded that NCS-1 in low micromolar concentrations effectively binds negatively charged phospholipids (as was correctly presumed from membrane binding studies) and the binding is therefore consistent with a physiological interaction.

Localization of NCS-1 in living cells has been previously characterized in a number of works [PMID 11836243, 14505460, 14600268, 16053445]. However, there are some questions left after these interesting studies. For instance, it was demonstrated, that NCS-1 and another NCS protein hippocalcin display exactly the same subcellular localization, which would indicate their similar mechanisms of membrane interaction. However, the comparison of the previous data with the results of our study indicates that phosphoinositide preference and mechanism of phospholipid binding of these proteins are drastically different. Thus, our study being focused on determining structural basis underlying membrane binding phospholipid recognition by NCS-1 provide important new insight into fundamental mechanisms of regulatory specificity of NCS-1 and other NCS proteins, as well as completely new data pointing at highly specific interaction of NCS-1 with crucial signaling phospholipid PI3P and other phosphoinositides. We agree with the reviewer that NCS1 interaction with membranes in living cells is more differentiated mostly due to different distribution of phospholipids and the presence of cholesterol and integral proteins. Yet, this interaction can be in some way assessed by using crude membrane preparations as it was done in our work. While there are methods that could allow to confirm NCS-1 interaction with PI3P in living cells (for instance by using a fluorescent biosensor [PMID 16053445]), this subject is highly time-consuming (Biomolecules provides only 10 days for major revision) and. to our opinion, deserves a separate study.

The suggestion of a specific PI3P-binding site in the N-terminal region of NCS-1 is from modeling studies using unmyristoylated NMR structure only, and so lacks any experimental evidence. Whilst the prediction looks plausible the authors make significant conclusions about binding modes based on this. This needs to be tested experimentally as, with reference to the authors results, K7 would be an integral component of this PI3P-binding site, yet PIP-Strip results suggest that mutation of K7 has no significant affect on PI3P binding compared to WT. The conclusions suggested by the authors do not seem to be supported by the results.

ANSWER:

Indeed, we did not found any significant differences in PI3P binding to K7E mutant, which lucks one of the residues forming the PI3P-binding site predicted in silico. However, our modeling experiments imply that K32, as well as a network of hydrogen bonds, provided by S4, Q28 and Q29, can participate in PI3P binding and stabilize it even in the absence of K7. It is well recognized that the binding of proteins to rare phosphoinositides (such as PI3P), as compared to PS or major phosphoinositides, is less reliant on net charge as on the complex arrangement of hydrogen bonds in the site, restricting the choice of the ligand by number and position of phosphate groups [reviewed in PMID:12694559]. Thus, K7E mutation might not be detrimental to PI3P binding, but rather destabilizing enough that the other phosphoinositides (namely PI4P, PI5P and PI(3,4)P2) would fit in better. This is exactly what we saw in our experiments as K7E mutation decreases selectivity towards PI3P as compared to other phosphoinositides (see Figure 5c in the updated manuscript).

In general, we totally agree with the reviewer that the interaction of PI3P with the described specific N-terminal site in NCS-1 needs to be verified in direct experiments. However, in order to reliably confirm such complex multipoint (structural) site one might use triple/quadruple mutants as single substitutions may not significantly affect the observed high-affinity interaction (please see above). Meanwhile, in contrast to N-terminal lysines addressed in this study, the other residues forming PI3P-bindimng site (Q28, Q29 and K32) are located in the protein core and their substitution (especially yielding triple or quadruple mutants with K7 substitution) may indirectly influence PI3P binding by affecting overall structure of the protein. Thus, the rigorous proof of the site requires structural and functional characterization of multiple mutants, which requires more time and, to our opinion, deserves a separate study.

With respect to the first binding mode, myristoylated NCS-1 is shown to bind to membranes (Fig.1) and the subsequent experiments are conducted with unmyristoylated protein with reference only to a similar WT binding preference on PIP-Strips. The binding mutants were not tested within this context, and there seems to be no binding to PS on the PIP-Strips to warrant the quantification.

ANSWER:

To address this concern we performed additional liposome-binding experiments (please see our response to the Comment 2), confirming that unmyristoylated and myristoylated NCS-1 display the same phospholipid binding patterns. This was necessary as we faced methodological limitation: our N-terminal mutants displayed hindered myristoylation due to critical alteration of their myristoylation motif by the introduced substitutions (especially by K7E substitution) [PMID:11955007]. Yet, given that the effect of phospholipid content on membrane binding of NCS-1 is better seen in the absence of myristoyl group (please see Figure 1 in the updated manuscript) and that both forms of the protein display similar phospholipid binding patterns (please see Figure 2 in the updated manuscript), unmyristoylated forms of NCS-1 mutants were reasonably used in the subsequent studies.

We agree with the reviewer that PS binding on dot-blot membranes is weaker than phosphoinositide binding, but it is still considerably more prominent not only compared to blank, but also compared to PC and PE and even some phosphoinositides (PI(3,4)P2). The observed level of staining is recognized as significant both by the manufacturer and by other researchers (exemplified in [PMID 25201878, 16053445, 14662869, 15240152]). It should be noted that the resulting plot in Figure 5b is based on the data from several dot-blot experiments and we can provide for the Reviewer the strip fragments from one of the parallel experiment, which are more illustrative in respect to PS binding (please find an attached .doc file). We indeed realize that the results of PIP strips binding may not be directly transferrable to full membranes or liposomes, as the strips contain membrane monolayers instead of bilayers (therefore the study was complemented by liposome-binding experimants as the Reviewer suggested) and therefore the binding is generally weaker. Yet, this means that any difference detected even in PIP strips experiments is indeed reliable.

Additionally the binding of unmyristoylated NCS-1 mutants to membranes (Fig 3) is shown to be independent of Ca2+ in photoreceptor membranes – however there does seem to be significant differences without Ca2+ for some mutants in hippocampal membranes and this needs to be explained further. The choice of t-test for multiple comparisons is not appropriate.

ANSWER:

The interaction of unmyristoylated NCS-1 with hippocampal membranes is generally very weak and the effects of the mutations are indistinguishable regardless the presence of calcium. In particular, the binding of all mutants is within the error range determined for the binding of wild type NCS-1 both in the absence and in the presence of calcium. Thus, the apparent Ca2+-sensitivity of the mutants mentioned by the Reviewer is not discussed in the manuscript, as the signal/noise ratio in these experiments is too low. We totally agree with the Reviewer that the choice of t-test for multiple comparisons is not appropriate in this case. Therefore, we recounted statistical significance using Mann-Whitney U test, which is more feasible for small datasets. According to these new estimations, no difference was found between the binding to hippocampal membrane of wild type NCS-1 and the mutants regardless the presence of calcium. Thus, in the updated manuscript the asterisks are removed from the bars corresponding to K3,7E and K3,7,9E mutants (Figure 4b).

Reviewer 3 Report

The submitted manuscript entitled “Membrane binding of Neuronal Calcium Sensor-1: highly specific interaction with phosphatidylinositol-3-phosphate” by Viktoriia Baksheeva , Ekaterina Nemashkalova , Arthur Zalevsky , Vasily Vladimirov , Natalia Tikhomirova , Pavel Philippov , Andrey Zamyatnin Jr. , Dmitry Zinchenko , Sergey Permyakov and Evgeni Zernii addresses potential membrane interactions of the calcium sensor NCS-1. The authors have utilised a number of N-terminal mutants of the sensor to try to define membrane-binding mechanisms, which Is coupled with an in vitro approach to demonstrate lipid interactions. Overall the manuscript is well presented, clearly written, and the experimental research is clearly defined. However, in my opinion, although the evidence of phosphatidylinositol monophosphate binding is a novel and well supported observation, the work is at a very preliminary stage and would not warrant publication in this journal without significant further supportive evidence.

A number of major concerns are raised which would have to be significantly addressed before consideration for publication, specifically;

The evidence for binding mechanisms presented is based solely on in vitro observations using recombinant protein. These observations also rely on crude membrane preparations from neuronal sources to monitor bound fractions, and also adsorbed lipid binding assays to suggest binding specificity. As the authors discuss the relative lipid compositions of the total cellular membrane preparations at length, and suggest differences in these relate to the protein binding modality, this does not account for differences in lipid compartmentalization and enrichment in the intact cells. Where a crude membrane preparation has a lower lipid composition, this may not be relevant if the composition of a targeted internal lipid compartment is enriched to a significant extent. Analysis within a cell-based context may add supporting evidence, even an over-expressed system utilizing the mutant forms the authors have. Furthermore the use of artificial membranes or liposome/micelle-bound fractions would give better evidence of association. The suggestion of a specific PI3P-binding site in the N-terminal region of NCS-1 is from modeling studies using unmyristoylated NMR structure only, and so lacks any experimental evidence. Whilst the prediction looks plausible the authors make significant conclusions about binding modes based on this. This needs to be tested experimentally as, with reference to the authors results, K7 would be an integral component of this PI3P-binding site, yet PIP-Strip results suggest that mutation of K7 has no significant affect on PI3P binding compared to WT. The conclusions suggested by the authors do not seem to be supported by the results. With respect to the first binding mode, myristoylated NCS-1 is shown to bind to membranes (Fig.1) and the subsequent experiments are conducted with unmyristoylated protein with reference only to a similar WT binding preference on PIP-Strips. The binding mutants were not tested within this context, and there seems to be no binding to PS on the PIP-Strips to warrant the quantification. Additionally the binding of unmyristoylated NCS-1 mutants to membranes (Fig 3) is shown to be independent of Ca2+ in photoreceptor membranes – however there does seem to be significant differences without Ca2+ for some mutants in hippocampal membranes and this needs to be explained further. The choice of t-test for multiple comparisons is not appropriate.

The submitted manuscript highlights an interesting observation that will be of interest in this field, and should at some point be published. However in the present form would require significant additional experimental work.

Author Response

This study addresses the mechanism of interaction of NCS-1 to membranes and demonstrates different mode of interaction including electrostatic-mediated binding. The study is well performed overall. However, there are a few issues that should therefore be addressed. The manuscript is overall well written. However definite article are sometimes missing throughout the manuscript. This needs to be checked.

ANSWER:

We truly appreciate the reviewer’s thoughtful analysis of our study. Please, find our step-by-step response below. For the convenience, the introduced changes are highlighted in the updated manuscript with blue font.

Line 104: indicate lysine in “the residues”

ANSWER:

The correction has been introduced into the manuscript.

Lines 235-6. The following sentence is unclear: “Given that the effect of phospholipid content on membrane binding of NCS-1 is better seen in the absence of myristoyl group”. Please rephrase and clarify.

ANSWER:

The statement is corrected.

Section 3.4. The authors claim that unmyristoylated NCS-1 binds to PA but this is not evident from the lipid blot (a) and the quantification (b) shows an overall low level. The statement should at least be toned-down in relation to the interaction signal given by PS.

ANSWER:

Indeed, the binding to PA for the myristoylated protein is significantly weaker compared to PS. The Results section was updated to better reflect this observation.

In addition, the importance given to PS should also be toned down. The evidence is not strong in particular from the lipid blot (a), showing the comparison of the WT to the triple mutant. In addition, the graph appears to show a greater decrease with the triple mutant in the graph compared to the blot. This needs to be addressed.

ANSWER:

We agree with the reviewer that PS binding on dot-blot membranes is weaker than phosphoinositide binding, but it is still considerably more prominent not only compared to blank, but also compared to PC and PE and even some phosphoinositides (PI(3,4)P2). The observed level of staining is recognized as significant both by the manufacturer and by other researchers (exemplified in [PMID 25201878, 16053445, 14662869, 15240152]). Furthermore, during the revision, we performed a series of additional experiments to characterize NCS-1 interaction with PS, PE, PC and PI-containing liposomes (please, see Section 3.2 in the updated manuscript). It was shown that NCS-1 indeed displays preference to PS over PI, PE and PC, regardless of myristoylation. As for the effect of triple charge inversion, it was identified on the basis of several dot-blot experiments with the subtraction of blank, and statistical calculations recognized it as significant. The resulting plot in Figure 5b is based on the data from several dot-blot experiments and we can provide for the Reviewer the strip fragments from one of the parallel experiment, which are more illustrative in respect to PS binding to WT NCS-1 and its mutant K3,7,9E (please find the attached .doc file).

Figure 4. Please show a separate quantification graph (from c) for PI3P for the WT and different mutants.

ANSWER:

The figure is updated with the inset plot, representing relative efficiency of PI3P binding by NCS-1 and its mutants. The binding is normalized to the densitometric count of the blank spots to account for any procedure differences in ECL staining of the strips.

Figure 5: in line with the results suggested by the docking experiment, it would be useful to mutate K32 with or without mutating K7 and test these mutants on lipid blots. K3 and K9 could also be tested.

ANSWER:

We totally agree with the reviewer that the interaction of PI3P with the described specific N-terminal site in NCS-1 needs to be verified in direct experiments. In fact, in order to reliably confirm such complex multipoint (structural) site one might use triple/quadruple mutants as single substitutions may not significantly affect the observed high-affinity interaction. Consistently, we did not found any significant differences in PI3P binding to K7E mutant lucking one of the residues forming the site predicted in silico. From the other side, in contrast to N-terminal lysines addressed in this study, the residues Q28, Q29 or K32 are located in the protein core and their substitution (especially yielding triple or quadruple mutants with K7 substitution) may indirectly influence PI3P binding by affecting overall structure of the protein. Thus, the rigorous proof of the site requires structural and functional characterization of multiple mutants, which is a time-consuming task (Biomolecules provides only 10 days for major revision) and, to our opinion, deserves a separate study.

Figure 4: K3 and K9 have the best effect on PI3P binding and the side chain should be indicated in the 3D structure to show their orientation in relation to the proposed orientation of the lipid, as in Figure 5b. Could these two lysines promote interaction, perhaps in combination with K32?

ANSWER:

The careful processing of the results of several parallel PIP strips experiments (including normalization of the densitometric count of the spots to each blank) reveled that there are no statistically significant differences between the interaction of K3E, K9E and other NCS-1 forms with PI3P (please see the inset in Figure 5c in the updated manuscript). Yet, we modified Figure 5 (Figure 6 in the updated manuscript) in order to include K3 and K9 as the Reviewer suggested. Now it shows that these residues do not interact with PI3P according to our model.

Discussion. The known mechanism of interaction of PS to other proteins should be discussed as well.

The Discussion is improved according to the Reviewer`s suggestion.

Round 2

Reviewer 1 Report

The revised manuscript addresses my previous concerns.

Author Response

We want to thank the reviewer for positive evaluation of our work.

Reviewer 2 Report

The authors have addressed all comments very thoroughly and have greatly improved relevant parts of the manuscript. This warrants publication.

Author Response

(The authors gave the same response as above.)

Reviewer 3 Report

The authors’ revised manuscript has addressed a number of the initial concerns that were raised previously. The addition of further experimental evidence for the interaction of NCS-1 with specific lipid species using a liposome-binding approach and FCS analysis enhances the paper. The authors’ responses to the other points (1-3) are generally well answered and I thank them for the additional explanatory dot blot provided. The further editing of the manuscript as requested by the other reviewer has also enhanced and explained results more clearly. Addressing the few points below, which may be done by editing the manuscript rather than inclusion of new data, would make the paper suitable for publication in my opinion.

Regarding the two main points for further evidence, namely the specific interaction of NCS-1 with PI3P in a cellular system and the experimental testing of the predictions inferred from the in silico putative binding site modelling, I understand that inclusion of this may be outside of the scope of this publication due the time involved in generating these data. However the authors should explain these limitations in the paper as together this lack of information impacts on the author’s conclusion that NCS-1 is involved in lipid signal transduction via PI3P. The possibility is not ruled out that specific binding of NCS-1 to phosphatidylinositol phosphates primarily is membrane localisation rather than assigned signalling function (unless some mechanistic evidence is presented). Other points are dealt with by reference to the answers provided;

ANSWER:

As it was suggested by the reviewer, we performed additional experiments (see Results section 3.2 and Methods section 2.6 in the updated manuscript) …. The results are presented in the new Figure 2a-c. According to these data, NCS-1 indeed demonstrates strong, predominantly Ca2+-independent interaction with preference to negatively charged phospholipids PS and PI over more abundant but zwitterionic/neutral PE and PC. The same pattern is retained for the unmyristoylated form of the protein, even though the overall intensity of binding is lower. These observations additionally justify using unmyristoylated forms of NCS-1 and its mutants in our subsequent experiments.

The new Fig. 2 shows the 10-fold weaker association of unmyristoylated NCS-1 very nicely with the liposomes. However I do not agree that the same pattern is retained for myristoylated and unmyristoylated NCS-1 (2a and 2c). The graph seems to suggest that unmyristoylated NCS-1 has a much lower binding efficiency for PI than the myristoylated form, and that this is now almost equivalent to affinity for PC and PE? Also in the absence of Ca2+ the affinity of unmyristoylated NCS-1 for PS also approaches these levels. NCS-1 would be myristoylated in cells, which would confer the PS/PI binding affinity. However the mutants that are generated and subsequently used are of the unmyristoylated form i.e. with reduced affinity for PI and PS (in the absence of Ca2+);

Do Fig 4a and 4b (approx. 50% less PI/PS in hippocampal membranes) indicate that K mutants that have a reduced membrane binding do so via interaction predominantly with PS? The unmyristoylated mutants have a higher affinity for PS than PI (Fig 2c). What are the relative ratios of PI and PS in these membranes? This would make sense if mode 1 binding is by –ve charge alone, enhanced by myristoylation (as either lipid species would provide this). This should be explained.

ANSWER:

We agree with the reviewer that PS binding on dot-blot membranes is weaker than phosphoinositide binding, but it is still considerably more prominent not only compared to blank, but also compared to PC and PE and even some phosphoinositides (PI(3,4)P2). The observed level of staining is recognized as significant both by the manufacturer and by other researchers (exemplified in [PMID 25201878, 16053445, 14662869, 15240152]). It should be noted that the resulting plot in Figure 5b is based on the data from several dot-blot experiments and we can provide for the Reviewer the strip fragments from one of the parallel experiment, which are more illustrative in respect to PS binding (please find an attached .doc file). We indeed realize that the results of PIP strips binding may not be directly transferrable to full membranes or liposomes, as the strips contain membrane monolayers instead of bilayers (therefore the study was complemented by liposome-binding experimants as the Reviewer suggested) and therefore the binding is generally weaker. Yet, this means that any difference detected even in PIP strips experiments is indeed reliable.

Care should be taken with comparison to the output from the dot blot strips. The monolayer/bilayer issue should not be relevant as mutant NCS-1 will only “see” one surface, although curvature could be an issue. It is good to see that the results are all normalised to the blank control for each blot as in the extra blots provided there is also a significant difference in the blank on each bot, generated by the ECL procedure. For Fig 5a; The PS and PI staining are very weak, which fits with the unmyristoylated protein. This is equivalent to the PE/PC staining which conforms with Fig 2c. This should be explained. Comparing the WT and myristoylated WT blots; shouldn’t the PI spot reflect a 20-fold higher binding and the PS spot reflect a 10-fold higher binding?

ANSWER:

To address this concern we performed additional liposome-binding experiments (please see our response to the Comment 2), confirming that unmyristoylated and myristoylated NCS-1 display the same phospholipid binding patterns. This was necessary as we faced methodological limitation: our N-terminal mutants displayed hindered myristoylation due to critical alteration of their myristoylation motif by the introduced substitutions (especially by K7E substitution) [PMID:11955007].

Does the fact that the K mutants could not be myristoylated reflect changes in the structure of NCS-1. Obviously the K7 mutation has impacted on the myristoylation site (motif or structure or both?). Also looking at the MST data in Fig 3, the K3,7E mutant displays a 6oC reduction in Tm, which could be conceived as a significant reduction in thermal stability. It is difficult to conclude from the increases seen in thermal stability but obviously the Tm is calculated from more components than just the stability of the native protein comparative to the unfolded state, so large differences suggest that some change is occurring. This should be revised in lines 315-317. Could recoverin be used as a control to test the phosphoinositide-specific binding site model? On the alignment (Fig 6c) this does not seem to have a conserved site and should therefore show less binding to PI3P in liposomes or strips?

Minor points;

Line 375 and Fig 5c; is it correct that mutants acquire the ability to bind to PI(3,4)P2? This species seems to indicate the lowest binding for most mutants. K3,7E has higher binding but the other mutants seem to have a marginal difference. Larger differences (compared to WT) are seen with PI(3,5)P2 and PI(4,5)P2?

Line 323; Figure 3a is now Figure 4a.

Line 387; do the authors mean inositol monophosphates? Should this be phosphatidylinositol phosphates (phosphatidylinositol monophosphates)?

Author Response

The authors’ revised manuscript has addressed a number of the initial concerns that were raised previously. The addition of further experimental evidence for the interaction of NCS-1 with specific lipid species using a liposome-binding approach and FCS analysis enhances the paper. The authors’ responses to the other points (1-3) are generally well answered and I thank them for the additional explanatory dot blot provided. The further editing of the manuscript as requested by the other reviewer has also enhanced and explained results more clearly. Addressing the few points below, which may be done by editing the manuscript rather than inclusion of new data, would make the paper suitable for publication in my opinion.

ANSWER:

We want to thank the reviewer for careful inspection and overall positive evaluation of our work. Please find below our step-by-step responses to the additional comments. For the convenience, all new corrections are highlighted in the updated manuscript in grey color.

Regarding the two main points for further evidence, namely the specific interaction of NCS-1 with PI3P in a cellular system and the experimental testing of the predictions inferred from the in silico putative binding site modelling, I understand that inclusion of this may be outside of the scope of this publication due the time involved in generating these data. However the authors should explain these limitations in the paper as together this lack of information impacts on the author’s conclusion that NCS-1 is involved in lipid signal transduction via PI3P. The possibility is not ruled out that specific binding of NCS-1 to phosphatidylinositol phosphates primarily is membrane localisation rather than assigned signalling function (unless some mechanistic evidence is presented).

ANSWER:

We totally agree with the reviewer that exact physiological function of NCS-1 complex with PI3P was not determined but only predicted based on the results of our work. Indeed, it cannot be ruled out that the revealed high-affinity binding of NCS-1 to phosphatidylinositol phosphates ensures its specific membrane localization rather than directly contributes to phosphoinositide-dependent signaling. For instance, in living cells, NCS-1, similarly to hippocalcin [PMID:16053445], may be predominantly associated with different phosphatidylinositol phosphates (including PI3P, PI5P, etc.), which may govern, for instance, its targeting to special sites on plasma or Golgi membranes. Such localization can, in turn, compartmentalize NCS-1 with its binding partners thereby maintain Ca2+-dependent regulatory function of the protein. Future studies are required to establish the exact physiological meaning of NCS-1 interaction with PI3P and the other phosphoinositides in different types of neurons. We think that the uncovering of this function as well as confirming the structure of the predicted site are quite challenging issues that might be resolved in further studies.

The respective reasoning, as well as the future directions of studies, including those suggested by the reviewer, are now introduced in the updated manuscript (please see page 16).

Other points are dealt with by reference to the answers provided;

The new Fig. 2 shows the 10-fold weaker association of unmyristoylated NCS-1 very nicely with the liposomes. However I do not agree that the same pattern is retained for myristoylated and unmyristoylated NCS-1 (2a and 2c). The graph seems to suggest that unmyristoylated NCS-1 has a much lower binding efficiency for PI than the myristoylated form, and that this is now almost equivalent to affinity for PC and PE? Also in the absence of Ca2+ the affinity of unmyristoylated NCS-1 for PS also approaches these levels. NCS-1 would be myristoylated in cells, which would confer the PS/PI binding affinity. However the mutants that are generated and subsequently used are of the unmyristoylated form i.e. with reduced affinity for PI and PS (in the absence of Ca2+);

Do Fig 4a and 4b (approx. 50% less PI/PS in hippocampal membranes) indicate that K mutants that have a reduced membrane binding do so via interaction predominantly with PS? The unmyristoylated mutants have a higher affinity for PS than PI (Fig 2c). What are the relative ratios of PI and PS in these membranes? This would make sense if mode 1 binding is by –ve charge alone, enhanced by myristoylation (as either lipid species would provide this). This should be explained.

ANSWER:

Indeed, the investigation employing structurally altered forms (s.a. mutants) of proteins always faces with some degree of approximation. However, although we were confined to using unmyristoylated proteins due to methodological reasons, we always tried to verify our conclusions considering the limitations of using these forms. Thus, the differences in PS affinities between myristoylated and unmiristoylated NCS-1 represent an interesting phenomenon, but it does not contribute to our interpretation of phospholipid preference of the mutants (Fig. 5) as it was studied only for Ca2+-loaded forms. In turn, the reduced binding of PI by unmyristoylated protein may generally reflect only on NCS-1 binding to native membranes presented on Fig. 4. However, this is also not the case, since, as it was correctly noted by the reviewer, the observed effects of the mutants are predominantly associated with altered interaction with PS.  Indeed, photoreceptor membranes contain up to 24% of PS but very low (<2%) concentration of PI, whereas hippocampal membranes contain low concentrations of both PS (<7%) and PI (<4%). Thus, the unmyristoylated NCS-1 displays higher affinity to photoreceptor membranes due to relatively high amount of PS in them and this binding becomes expectedly reduced upon the charge inversion in N-terminus of the protein. By contrast, the binding to hippocampal membranes is initially lower and insensitive to the mutations. It should be noted that such properties might be similarly peculiar to myristoylated forms of the protein. Although mNCS-1 exhibits higher affinity to PI, the contribution of this binding to the total interaction with photoreceptor and hippocampal membranes would be unessential due to low content of this phospholipid in both membranes.     

Nevertheless, the high-affinity binding of myristoylated (i.e. native-like) NCS-1 to PI represents an important novel feature of the protein, which could be relevant to other cellular systems. We totally agree with the reviewer that it is the “mode 1 binding” that is enhanced in the presence of myristoyl group (actually, this can be seen from comparison of the signal magnitudes in Figures 2A and 2C). This makes perfect sense since myristoyl group being inserted in the membrane may stabilize protein in position, where its N-terminus contacts with the charged surface of bilayer (in fact, such configuration is demonstrated for recoverin by solid phase NMR [PMID:12767213]). Given that PS and PI differ in size and charge localization, one can assume that the presence of myristoyl group could promote the interaction of NCS-1 with PI more efficiently than with PS, thereby equalizing total binding of these phospholipids (Figure 2A versus Figure 2C).

The respective reasoning is now introduced in the updated manuscript (please see page 13).

Care should be taken with comparison to the output from the dot blot strips. The monolayer/bilayer issue should not be relevant as mutant NCS-1 will only “see” one surface, although curvature could be an issue. It is good to see that the results are all normalised to the blank control for each blot as in the extra blots provided there is also a significant difference in the blank on each bot, generated by the ECL procedure. For Fig 5a; The PS and PI staining are very weak, which fits with the unmyristoylated protein. This is equivalent to the PE/PC staining which conforms with Fig 2c. This should be explained. Comparing the WT and myristoylated WT blots; shouldn’t the PI spot reflect a 20-fold higher binding and the PS spot reflect a 10-fold higher binding?

ANSWER:

We agree that PIP strips assay has many limitations and its results should be interpreted cautiously. Nevertheless, we would like to note that the respective experiments were performed in repetitions and we are confident of revealed tendencies. In our presentation in Fig. 5 we tried to focus on two things. Firstly, it is high affinity of NCS-1 to PI3P, which is generally unaffected by the mutations. To our opinion, this observation is best seen form one of the actual strips (raw data) given in Fig. 5a. Secondly, it is weaker but still significant binding of NCS-1 to PS, which, in contrast, is affected by the mutations even in PIP strips assay. This second effect is better seen form the other strips (one of them was attached to our previous Response to Reviewers), but we decided to demonstrate it as error bars generated on the base of the results of several experiments (Fig. 5b) thereby emphasizing its reproducibility. Importantly, in these experiments we noticed the same trend, namely PS binding > PI binding (as well as PE/PC binding), as was observed for the unmyristoylated protein in the liposome-binding assay (Fig. 2C), which confirms relevance of the both methods.

The absence of 10/20-fold increase in PS/PI binding monitored by PIP strips assay is not surprising for us as we think that myristoyl group cannot enhance protein binding to monolayer as effectively, as in the case of a membrane bilayer. Yet, this makes PIP strips even more suitable for studying phospholipid preference of the acylated proteins, because in this case the interaction is more governed by a specific protein site rather than by fatty acid moiety. In fact, as it is stated on page 11 (line 380), the general aim of the respective experiment was not comparing the binding of two NCS-1 forms quantitatively (this was done in the liposome-binding assay) but demonstrating that high-affinity binding of PI3P is myristate-independent and, thereby, might occur via a specific site (binding mode 2). In our opinion, this aim was attained successfully.

Does the fact that the K mutants could not be myristoylated reflect changes in the structure of NCS-1. Obviously the K7 mutation has impacted on the myristoylation site (motif or structure or both?). Also looking at the MST data in Fig 3, the K3,7E mutant displays a 6oC reduction in Tm, which could be conceived as a significant reduction in thermal stability. It is difficult to conclude from the increases seen in thermal stability but obviously the Tm is calculated from more components than just the stability of the native protein comparative to the unfolded state, so large differences suggest that some change is occurring. This should be revised in lines 315-317.

ANSWER:

Indeed, there was a moderate decrease in thermal stability of NCS-1 upon introduction of K3,7E substitutions. However, we did not observe such tendency in the case of K7,9E mutant or even triple mutant K3,7,9E exhibiting the major effects in phospholipid/membrane-binding assays (Fig. 4a, Fig. 5b). Furthermore, as it is stated in the manuscript (page 9), the alterations in Tm are peculiar only to Ca2+-free forms of the protein (apo-froms of NCS proteins are generally characterized by much less thermal stability than their Ca2+-bound conformers [PMID:24359287,    25714968, 21169352]), whereas the presence of calcium shifts melting point of all NCS-1 forms to >95oC indicating that all variants of the protein including K3,7E still bind Ca2+ and their core structure is largely unaffected by the mutations. Finally, all mutants retain Ca2+-dependent shift in SDS-PAGE mobility, which is characteristic of wild type NCS-1 and may serve as an intrinsic control of their structural integrity. It should be noted also that although K3,7E mutant displayed reduced stability in the absence of calcium, we did not observe any specific effects of this form in our assays, which stand out from overall tendencies. Nevertheless, we agree with the reviewer that the effects of the mutations on NCS-1 thermal stability in the Results section need to be defined more accurately (please see page 9).

According to the literature, effective myristoylation is achieved in proteins, where six N-terminal positions (residues 2-7, methionine is removed for myristoylation) are occupied by certain residues arranged in a specific order to ensure the efficient binding in the conserved substrate-binding site of NMTs [PMID:11955007]. In particular, the 7th position of such protein should be represented by a positively charged residue forming a charge interaction with aspartate-417 of yeast NMT-1. In our case, this residue is substituted by negatively charged glutamate. Thus, although we cannot exclude some complex structural effects, it is the charge repulsion introduced by 7E mutation that most likely makes NMT-1 binding (and thereby myristoyaltion) unfavorable. This suggestion is confirmed by our observation that only the mutants containing 7E substitution were completely unmodified, whereas in case of the other mutants (such as, for instance, K3,9E) myristoylated forms could be obtained.       

Could recoverin be used as a control to test the phosphoinositide-specific binding site model? On the alignment (Fig 6c) this does not seem to have a conserved site and should therefore show less binding to PI3P in liposomes or strips?

ANSWER:

We want to thank the reviewer for this thoughtful suggestion. Recoverin may certainly be used in future studies as a control to test the specificity of the predicted PI3P-binding site. Indeed, recoverin represents highly specialized, retina-specific NCS protein and the structure of its N-terminal region might reflect this specialization. While there is evidence of recoverin interaction with PS monolayers, it displays no affinity to PI [PMID: 27689444], which is almost absent in photoreceptor membranes. Comparison of phospholipid/phosphoinositide preference of recoverin to that of more prevalent in central nervous system NCS-1, neurocalcin, or visinin-like protein-1 could provide interesting insight into the mechanisms underlying functional diversity of NCS proteins.

Minor points;

Line 375 and Fig 5c; is it correct that mutants acquire the ability to bind to PI(3,4)P2? This species seems to indicate the lowest binding for most mutants. K3,7E has higher binding but the other mutants seem to have a marginal difference. Larger differences (compared to WT) are seen with PI(3,5)P2 and PI(4,5)P2?

ANSWER:

We are appreciated to the reviewer for this notion. Indeed, double and triple mutants acquired ability to bind PI(3,4)P2. However this effect is much lower than the effect of the mutations on NCS-1 binding to PI(3,5)P2, which is now emphasized in the updated manuscript (please, see page 11). 

Line 323; Figure 3a is now Figure 4a.

ANSWER:

The mistake is corrected.

Line 387; do the authors mean inositol monophosphates? Should this be phosphatidylinositol phosphates (phosphatidylinositol monophosphates)?

ANSWER:

The mistake is corrected.